# Simulating the Environmental Spread of SARS-CoV-2 via Cough and the Effect of Personal Mitigations

**DOI:** 10.3390/microorganisms10112241

**Published:** 2022-11-12

**Authors:** Claire Bailey, Paul Johnson, Josh Moran, Iwona Rosa, Jodi Brookes, Samantha Hall, Brian Crook

**Affiliations:** Health and Safety Executive Science and Research Centre, Harpur Hill, Buxton, Derbyshire SK17 9JN, UK

**Keywords:** cough, droplets, virus transmission, COVID-19, mitigation

## Abstract

Background: A cough is known to transmit an aerosol cloud up to 2 m. During the COVID-19 pandemic of 2020 the United Kingdom’s National Health Service (NHS), other UK government agencies and the World Health Organization (WHO) advised people to cough into their elbows. It was thought that this would reduce viral spread and protect the public. However, there is limited peer reviewed evidence to support this. Objectives: To determine if cough related interventions reduce environmental contamination, protecting members of the public from infection. Methods: Scientists and engineers at the Health and Safety Executive (HSE) laboratory used a human cough simulator that provided a standardised cough challenge using a solution of simulated saliva and a SARS-CoV-2 surrogate virus; Phi6. *Pseudomonas syringae* settle plates were used to detect viable Phi6 virus following a simulated cough into a 4 × 4 m test chamber. The unimpeded pattern of contamination was compared to that when a hand or elbow was placed over the mouth during the cough. High speed back-lit video was also taken to visualise the aerosol dispersion. Results and Discussion: Viable virus spread up to 2 m from the origin of the cough outwards in a cloud. Recommended interventions, such as putting a hand or elbow in front of the mouth changed the pattern of cough aerosol dispersion. A hand deflected the cough to the side, protecting those in front from exposure, however it did not prevent environmental contamination. It also allowed for viral transfer from the hand to surfaces such as door handles. A balled fist in front of the mouth did not deflect the cough. Putting an elbow in front of the mouth deflected the aerosol cloud to above and below the elbow, but would not have protected any individuals standing in front. However, if the person coughed into a sleeved elbow more of the aerosol seemed to be absorbed. Coughing into a bare elbow still allowed for transfer to the environment if people touched the inside of their elbow soon after coughing. Conclusions: Interventions can change the environmental contamination pattern resulting from a human cough but may not reduce it greatly.

## 1. Introduction

Coughing is well recognised as a means of spread of airborne transmissible infectious diseases and it therefore contributes to transmission in the indoor evironment. A persistent cough is one of the main symptoms of SARS-CoV-2 (COVID-19) infection [1], and at the height of the COVID-19 pandemic was one of the factors used to exclude people from work. This was complicated by evidence that emerged at the beginning of the pandemic that significant numbers of people could be asymptomatic carriers and shedders of COVID-19 [2]. Evidence from tightly defined and isolated populations, such as the passengers on the Diamond Princess cruise ship [3,4], showed that the proportion of asymptomatic carriers in the general public may be significantly underestimated [5], but infectivity was similar to that of those showing symptoms. A person may also shed the virus in a two day window before overt symptoms [6].

As well as isolation or exclusion, another means of preventing COVID-19 transmission that was applied in the workplace was social distancing [7,8]. This ranged from 1 m to 2 m [9], according to nationally and locally applied rules, which also changed at different times throughout the pandemic and depended on other mitigations put in place such as enhanced ventilation [10]. However, these recommendations were based on limited scientific data and to some extent influenced by what was practical and achievable. Some evidence suggested that, based on droplet size and emission dynamics, the virus could travel much further [11].

Recommended mitigations to limit droplet spread at a practical, personal level were to cough into hands or a tissue, or into the crook of the elbow, i.e., the inner elbow [12,13,14]. There is limited data on how effective these actions are, and also on the role that coughing into hands could then have on subsequent hand to surface transfer in terms of transmission dynamics. Robust data will provide businesses and industry with an evidence base on which to provide guidance to workers as they continue to repopulate offices and meeting rooms.

HSE scientists developed a cough simulation device as part of a project to test the protective effect of face shields [15]. It was designed to mimic the physical parameters of a human cough, enabling either an ultraviolet (UV) fluorochrome or a biological agent to simulate SARS-CoV-2 in aqueous suspension to be delivered as droplets in a size range comparable to a cough. This paper describes how further use of this cough simulator provided the opportunity to investigate how far droplets or particles from a cough can spread under defined conditions, how this can be mitigated and how this might impact on surface contamination.

## 2. Materials and Methods

### 2.1. Cough Simulator and Experimental Set Up

A cough simulator used previously [15] was adapted from an existing design [16] based on flow rate measurements of coughs from 47 human subjects with influenza [17]. The simulator comprised a ‘drive cylinder’ that ejected 4.2 L of air from a ‘lung cylinder’ through a ‘mouth’ outlet (Figure 1). The experimental set up for the cough simulator is shown in Figure 2. The flow rate against time matched the target profile of the original cough simulator [16]. The outlet was connected perpendicularly to a plastic pipe (1.2 m length × 0.04 m diameter). A pressurised airbrush (Badger 200; Badger Air-brush Co., Franklin Park, IL, USA) was used to spray an aqueous solution of a UV fluorochrome (1% Invisible Red (Chemox Pound, Farnborough, UK)) and/or bacteriophage into the pipe. Once the pipe was fully charged with spray from the airbrush the cough was initiated. The simulated cough was directed into test room.

The floor of a wooden test room with internal dimensions of 3 m H × 4 m W × 4 m D was lined with non-fluorescent black plastic sheet to enhance UV visualisation, then overlaid with a replaceable layer of clear plastic sheet. Yellow electrical tape was used to mark parallel lines at 0.5 m intervals from the cough origin and additional marks on each line to show the intersection of 10, 20, 30, 40 and 45 degree angles radiating out from the centre line from the initial cough’s origin (Figure 3). The cough simulator outlet pipe was inserted through a standard manikin head (‘Sheffield head’ (Inspec International Ltd., Salford, UK)) to deliver the cough from the centre line at 0 m. Mechanical ventilation to the room was switched off and doors to the room were partially closed to create near-still air conditions and minimise external interferences. It was not possible to fully shut the door due to the need to position the cough machine adjacent to the test area.

### 2.2. SARS-CoV-2 Simulant and Experimental Runs

The bacteriophage Phi6 (culture collection ref DSM 21518) was chosen, this being used as an airborne transmission simulant for SARS-CoV-2 in previous work [18]. A literature search determined a number of possible recipes for simulated saliva and preliminary tests (results not shown). One of these was selected based on its ability to maximise survival of viable Phi6 bacteriophage, without altering the anticipated spread pattern from the cough machine. 

Phi6 was grown in an 18 h culture of its bacterial host *Pseudomonas syringae* (culture collection ref DSM 21482) in a shaking incubator at 25 °C overnight. This was then centrifuged at 1690 g to remove cell debris and filter-sterilised using both a 0.45 µm filter and a 0.2 µm filter (Merck Millipore, UK) to remove any bacterial debris present. The resulting solution was then used in the testing.

In initial experimental runs, Phi6 was suspended in tryptone soya broth at a concentration of 1 × 10^9^ plaque forming units (PFU)/mL, this having been determined from literature to match the estimated SARS-CoV-2 viral load [19]. For later runs, the concentration was increased to 1 × 10^10^ PFU/mL to maximise detectable numbers and increase accuracy of the experiments. Phi6 suspensions were mixed in equal volumes with double strength simulated saliva solution (Table 1) [20] to achieve the working strength for the tests. The surface tension of the solution was measured using a torsion balance (Model OS (Torsion Balance Supplies, UK)).

Tryptone Soy Agar (TSA, Oxoid (Oxoid Ltd., Basingstoke, UK)) plates were spread with 300 µL of an 18 h culture of *Pseudomonas syringae* and placed at 83 pre-determined locations on the floor grid (Figure 3 and Table 2). An additional plate was also placed on the wall 4 m away from and facing the cough at head height.

Each test run comprised three coughs performed in quick succession at intervals of approximately one minute, this being the shortest turnround to allow the simulator to be re-primed. Three coughs was considered to simulate the natural cough process as it is unlikely that individuals would cough once only. This had been determined from a previous study [21] as delivering a measurable quantity of cough droplets. On completion of the test run the room was closed and the plates left for 10 min to allow the particles suspended in the air to settle. Initial experiments showed no appreciable increase in numbers settling on plates left longer than 10 min and up to 1 h, therefore, the shorter time scale was chosen for future tests as the more practical option. After 10 min the room was mechanically ventilated for 30 s to remove any fine particles still suspended in the air, and the lids were replaced on the plates as they were retrieved. 

Exposed TSA plates were incubated at 25 °C overnight. Clear halos in the resulting lawn of bacterial growth showed where a bacteriophage particle had landed and these were enumerated.

Three sets of tests were conducted to examine the effect of mitigation using a hand or elbow to contain the spread of a cough across the test room, as follows:Test A: Cough spread into the test room with no intervention.Test B: Cough spread into the room with a cupped human hand placed in front of the manikin mouth as a person would (See Figure 4).Test C: Cough spread into the room with the sleeveless human inner elbow placed in front of the manikin mouth as a person would (not touching the mouth).

### 2.3. Backlit Photography of the Coughs

A Phantom highspeed camera was used to capture the slow-motion images of the coughs, the camera was set to record an image every millisecond with each frame being exposed for 600 µs. Two 12,000 lumen lights were used to illuminate the scene, one of which was positioned behind the cough simulator to aid in the visualisation of the particles. The second light illuminated the front of the simulator.

The three different scenarios used within the environmental test were visualised. In addition, members of the public have also been observed to practice other interventions, such as suppression of a cough using a balled fist, a sleeved elbow or an elbow pushed more tightly to the mouth. These therefore were included in the study and analysed using backlit photography. 

### 2.4. Virus Survival and Fluorescence Visualisation of Environmental Transfer Following Interventions

A test rig was built comprising a flat wooden board (0.8 m × 0.3 m) onto which was mounted a 0.03 m diameter × 0.5 m long wooden pole. This was painted with black non-reflective paint as in Figure 5, and was used to mimic a handrail.

A 1% solution of Invisible Red (Chemox Pound, Farnborough, UK) was added to a Phi6 and simulated saliva solution so as to maintain the concentration of Phi6 used in previous tests. During method development it was determined that the addition of Fluorescent dye did not adversely affect viral viability and recovery. The fluorochrome allowed surface cross contamination to be visualised under UV light and photographed. The following series of test scenarios were conducted in duplicate to observe the effectiveness of mitigation using a hand or inner elbow placed in front of the manikin mouth to contain a cough;

Scenario 1: human hand placed in front of the manikin mouth, observed transfer to hand.Scenario 2: human hand placed in front of the manikin mouth, contact hand rail for three seconds and observed transfer to hand rail.Scenario 3: sleeveless human inner elbow placed in front of the manikin mouth, observed transfer to elbow.Scenario 4: sleeveless human inner elbow placed in front of the manikin mouth, place hand on inner elbow for three seconds (to mimic a person folding their arms) and hand on hand rail for three seconds. Observed transfer to both hand and hand rail.Scenario 5: as Scenario 3 but with sleeved arm; sleeved crook of a human elbow placed in front of the manikin mouth, observed transfer to elbow.Scenario 6: as Scenario 4 but with sleeved arm; sleeved human inner elbow placed in front of the manikin mouth, place hand on inner elbow for three seconds and hand on hand rail for three seconds. Observed transfer to both hand and hand rail.

Sterile pre-moistened sponge wipes (Sterile Sampling Sponge/Envirostik Kit, Technical Services Consultants Ltd., Heywood, Lancashire, UK) as employed in a previous study [22] were used to determine if viable virus was present from hand contact. The wipes were systematically rubbed across the test surface and placed back in the bag. 10 mL PBS was then added and massaged for 1 min to extract the virus. The resulting Phi6 suspension was added to 100 µL volumes to 3 mL Tryptone Soy agarose over lay containing 300 µL of actively growing 18 h culture of *Pseudomonas syringae*. The over lay was poured onto Tryptone soy agar and left to set before incubating at 25 °C for 12 h. Plaques were then counted and back calculated to estimate the number per swab. Before each run the hand, elbow and rail were disinfected to remove any microorganisms using disinfectant wipes (Distel, East Kilbride, UK).

## 3. Results and Discussion

### 3.1. Viral Particle Travel within the Environment

The surface tension of the simulated human saliva was determined as 0.058 M·m^−1^. This was comparable to previously reported experimental values [23,24].

The average environmental spread seen with an unmitigated cough (test A) was focussed on the centre line as shown in Figure 6A. There was observed to be an initial burst and elongated cloud with the highest concentration of virus (21 to 25 live viral particles) at the centre, 1 m away from the cough origin. Live viral particles were found to have spread to all parts of the test area but in lower numbers, with the exception of the settle plate at head height at 4 m from the cough origin. A slight drift to the right was observed, with slightly higher viral numbers down the right-hand side, probably due to slight air movement within the test room. Figure 6A shows the average cough pattern seen from three runs.

In test B, when a hand was placed in front of the outlet of the cough simulator, the pattern was more dispersed than the unmitigated cough, with 5 to 10 live viral particles at most locations throughout the room. The deposited viral particles contaminated plates further out into the room in slightly higher numbers than was seen for the unmitigated cough. This result is shown by the plate at 3 m and 20 degrees to the right showing an average of 16 to 20 viable counts over three runs (Figure 6B). An average of 3.33 viable viral particles was detected at head height at the back of the room, 4 m from the cough origin, highlighting the low level dispersal of viral particles throughout the room.

In test C, when an elbow was placed in front of the outlet of the cough simulator, the number of viable virus increased at 40 and 45 degrees from the central line from the cough origin. Contamination was also detected further away from the cough origin compared to both tests A and B, with up to 50 virus particles being detected at 3.5 m away from and to the left of the cough origin. In contrast to the unmitigated and hand tests the plates at the 4 m line also showed high numbers (40 to 45 viable viral particles). An average of 0.65 viable viral particles was detected at head height at the back of the room, 4 m from the cough origin. This was the result of 2 colonies on one of the three test runs. Figure 6C shows the average cough pattern seen from three runs.

### 3.2. Photographic Visualisation of the Cough Following Interventions

Backlit photographs clearly showed the particle cloud and its direction of travel. With the unmitigated cough, the cough travelled from the mouth and formed a cloud (Figure 7A). This elongated and gently dispersed. When a cupped hand was placed in front of the mouth (test B) the particle cloud was diverted from a forward plane and escaped the hand as a “star” pattern in a flat vertical plane (Figure 7B). A bare elbow (test C) was shown to divert the particle burst above and below the elbow, while still being propelled forward (Figure 6C). This cloud was seen to recombine as it travelled further from the cough origin.

Visualisation tests undertaken with a balled fist showed that it did not deflect the cough like the cupped hand over the mouth, but resulted in wider and less elongated cloud dissemination, (Figure 8).

Compared to intervention with a bare elbow, a sleeved elbow appears to entrap some of the particles thus reducing the particle cloud size and directing it in an upward manner (Figure 9A). An elbow pressed up close to the mouth or origin of the cough appears to visually reduce forward contamination compared to an unmitigated cough and the bare elbow (test C), deflecting much of it back towards the face of the person coughing. See Figure 9B.

### 3.3. Viral Survival and Transfer into the Environment Following Interventions

Fluorescence visualisation demonstrated that in all of the simulated scenarios saliva and bodily fluids were present on each surface contacted. This was consistent across both test runs performed and is shown in Figure 10A,B.

The numbers of virus retrieved per wipe sample are shown in Table 3. The results were lower than might be expected, based on the original titre used to prime the cough simulator. However, during initial method development the wipes were shown to have a consistent retrieval efficiency which, although only 40% of the expected total (data not shown), allowed direct and relative comparisons to be made across scenarios as shown in Table 3.

Live virus was detected on the hands and elbow before contact with the touch points, showing that virus was being expelled with the cough. It can be seen from visualisation of the fluorescent dye that the simulant body fluid was transferred to the hand rail and it was shown that some viral particles were transferred. However, the level of viral contamination was found to be low. Viable viral particles were still detectable on the hand after touching the handrail indicating that not all the virus was transferred to another surface. The surface wipes taken from the sleeved elbow showed low viral counts and no transfer to the hand or rail was detected in these scenarios. This suggests that the virus was likely entrapped/entrained within the material and therefore not easily transferred. However, it is also possible that the Phi6 did not survive the process of being transferred to the touch points during these tests. These results show the benefit of the dual approach of fluorescence visualisation along with the viral marker, as the fluorescence clearly shows transfer of fluid in all tests. However, this fluid may or may not contain live virus. 

## 4. Conclusions

This small-scale pilot study aimed to examine the effect of personal mitigations to prevent others from being exposed to a cough. It was shown that placing a hand or bare elbow over the mouth when coughing can deflect, but does not prevent, environmental exposure. The direction of the expelled cough was diverted from a frontal cloud to one that spread the contamination up and over an elbow placed in front of the mouth or in a flat sideways plane with a hand placed over the mouth. This means that, with a cupped hand, it is possible that those in front of the cough would have reduced exposure, however those to the side are potentially exposed to more viral particles than without mitigation/intervention. The photographic analysis indicated that a sleeved elbow may capture more of the aerosols and therefore would suggest that when coughing, an individual should cover their mouth with a sleeved elbow rather than their hand or bare elbow to reduce potentially exposing bystanders.

This study did not look at exposure that may occur to the rear of the cough due to restrictions in the test room. Some studies have indicated that cough mitigations, such as wearing a visor or mask may direct aerosols behind the individual coughing [25] and further research with the cough simulator would be beneficial, to compare this effect to frontal or sideward dissemination of the airborne virus when mitigation of the cough is caused by the hand or the elbow. 

The simulated cough was assumed to be representative of a real human cough and has been used in other research to mimic a cough. The particle size distribution was assumed to be representative of a human cough however the short explosive nature of the cough blast made measurement difficult [15]. 

This study showed that if a person coughed into their hand it is possible for live viral particles to be subsequently transferred to areas in the environment such as door handles. The advice given by the NHS, WHO and other public health bodies throughout the world during the COVID-19 pandemic was for a person to cough into their inner elbow. The rationale was that this would reduce environmental contamination as the hand was not contaminated. However, this study showed that, at least in the short term after coughing into an inner elbow, there is a need not to touch that site as it was possible to transfer live viral particles if a person crossed their arms, for example, thereby touching the cough site with a hand. It is clear from this study and the work of others that the general public should be mindful of viral transfer.

## Figures and Tables

**Figure 1 microorganisms-10-02241-f001:**
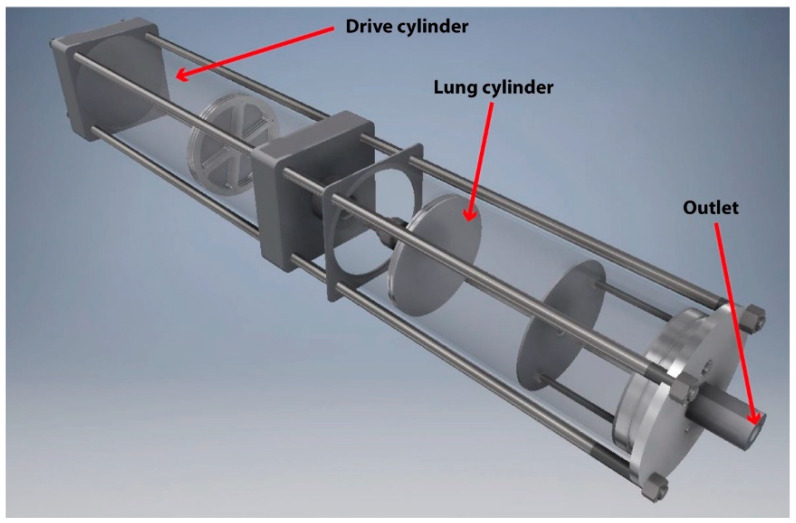
Design of the HSE cough simulator with the barrels transparent [15].

**Figure 2 microorganisms-10-02241-f002:**
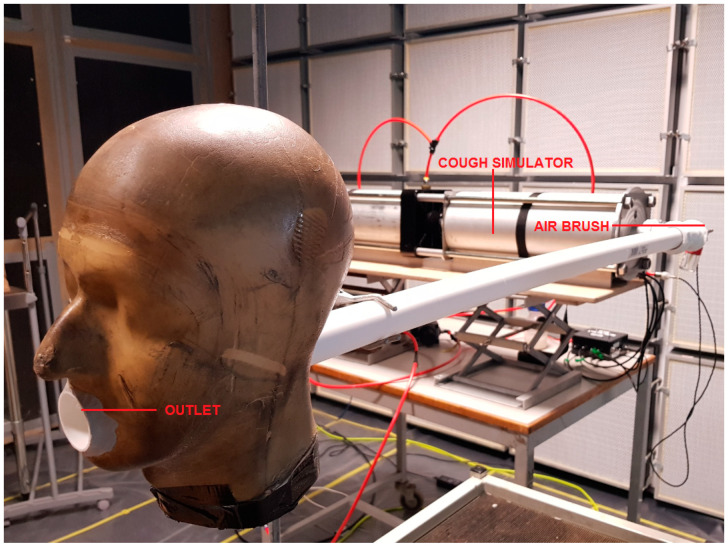
Experimental set up showing the cough simulator, airbrush and outlet pipe through a manikin head.

**Figure 3 microorganisms-10-02241-f003:**
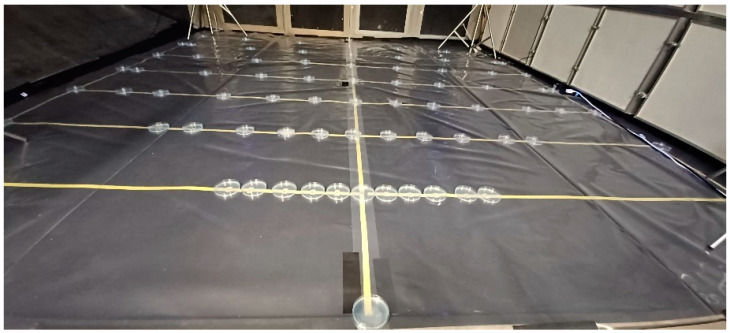
Test room with marked grid pattern and agar settle plates.

**Figure 4 microorganisms-10-02241-f004:**
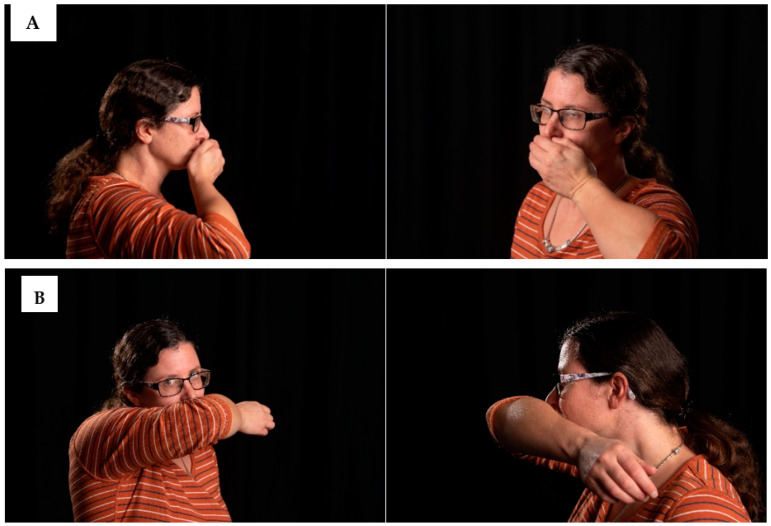
(**A**) shows an individual placing their hand over the mouth, at a distance that was typical of those we observed, and then used within the viral model. (**B**) shows and individual placing their elbow in front of the mouth, at a distance typical of those we observed and used within the tests.

**Figure 5 microorganisms-10-02241-f005:**
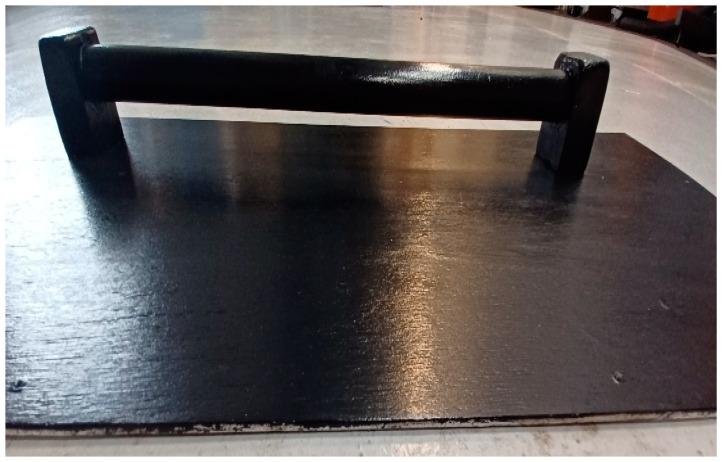
Board plate and handle used to mimic touch points.

**Figure 6 microorganisms-10-02241-f006:**
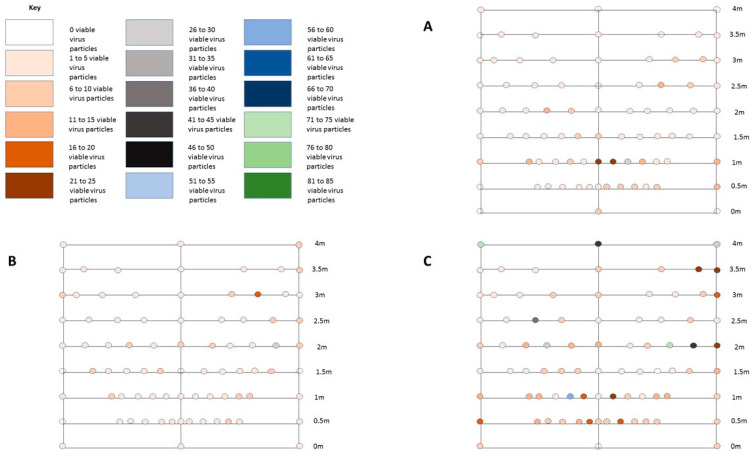
(**A**–**C**) Average environmental spread of cough (based on 3 tests) with no intervention (test **A**), cough with a hand cupped in front of the mouth (test **B**) and cough with inner elbow in front of the mouth (test **C**). All data points = average of three test runs.

**Figure 7 microorganisms-10-02241-f007:**
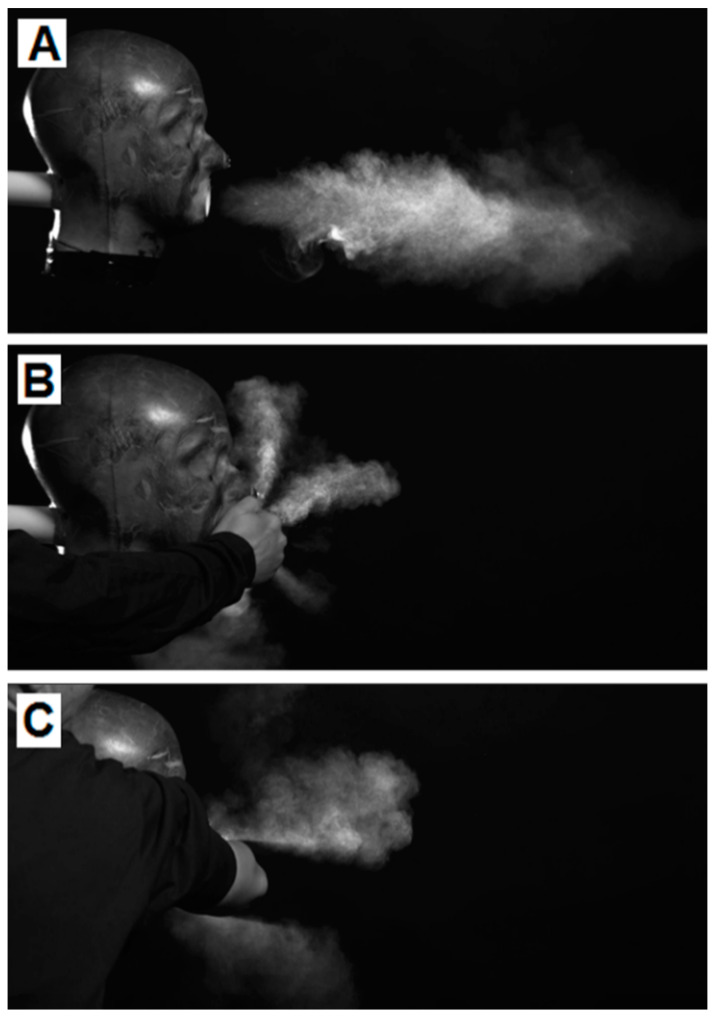
High-speed backlit photographs of test (**A**) the cough with no intervention, test (**B**) the cough with the hand over the mouth, test (**C**) the cough with a bare elbow.

**Figure 8 microorganisms-10-02241-f008:**
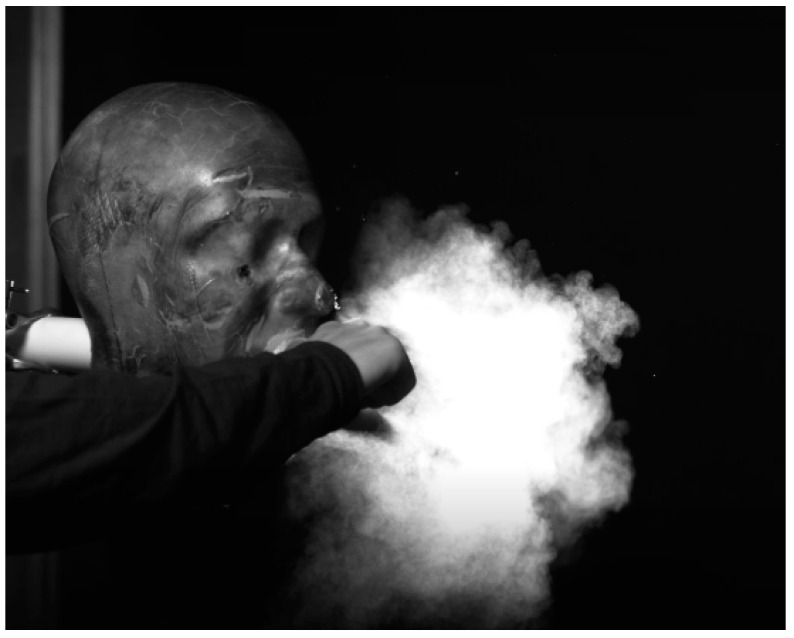
High-speed backlit photographs of the cough with balled fist.

**Figure 9 microorganisms-10-02241-f009:**
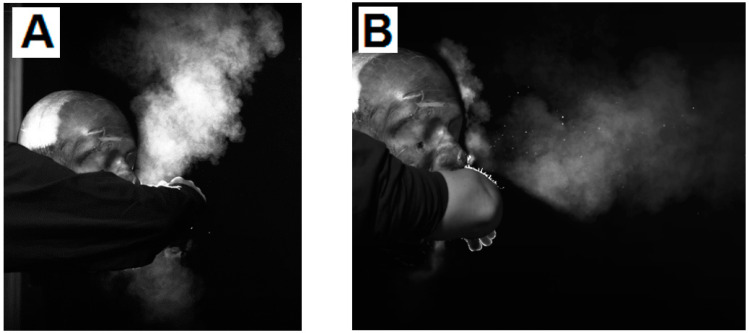
High-speed backlit photographs of the cough with (**A**) a sleeved elbow and (**B**) the elbow touching the mouth.

**Figure 10 microorganisms-10-02241-f010:**
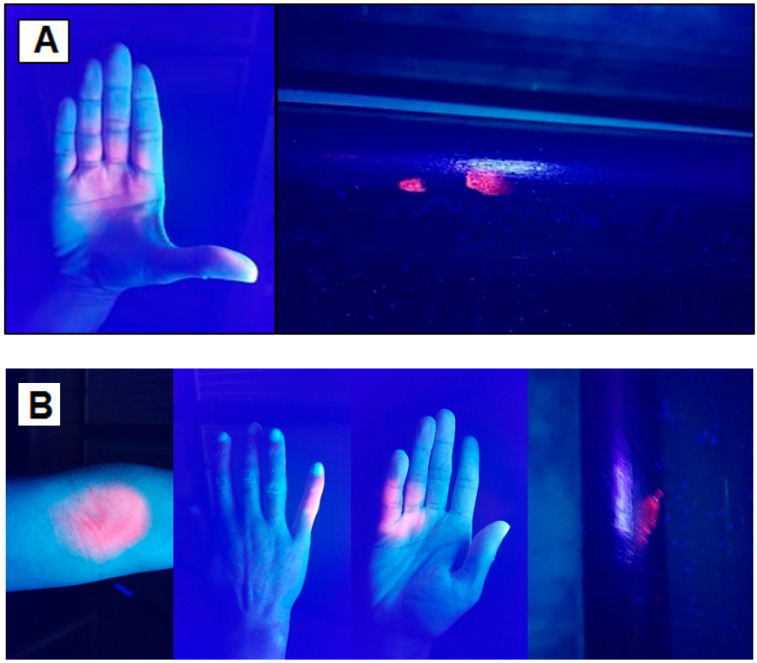
(**A**) A cough into a cupped hand and then contact with a door handle; (**B**) a cough in to a bare inner elbow, hand contamination from arms being crossed so that hand touches contaminated inner elbow and then contact with a door handle.

**Table 1 microorganisms-10-02241-t001:** Double strength Simulated Saliva recipe (used double strength stock to mix with equal amount of TSA broth and Phi6).

Component	Concentration (g/L)
Potassium Chloride	0.298
Sodium Chloride	0.234
Sodium Bicarbonate	4.2
α-Amylase (Porcine)	4.0
Mucin (Gastric)	2.0
Sterile distilled water	N/A

**Table 2 microorganisms-10-02241-t002:** Sample positions.

Sample Number	Description
1	0 m, outer edge on left
2	0 m centre
3	0 m, outer edge on right
4	0.5 m, outer edge on right
5	0.5 m, 45° degrees on left
6	0.5 m, 40° degrees on left
7	0.5 m, 30° degrees on left
8	0.5 m, 20° degrees on left
9	0.5 m, 10° degrees on left
10	0.5 m centre
11	0.5 m, 10° degrees on right
12	0.5 m, 20° degrees on right
13	0.5 m, 30° degrees on right
14	0.5 m, 40° degrees on right
15	0.5 m, 45° degrees on right
16	0.5 m, outer edge on right
17	1 m, outer edge on left
18	1 m, 45° degrees on left
19	1 m, 40° degrees on left
20	1 m, 30° degrees on left
21	1 m, 20° degrees on left
22	1 m, 10° degrees on left
23	1 m centre
24	1 m, 10° degrees on right
25	1 m, 20° degrees on right
26	1 m, 30° degrees on right
27	1 m, 40° degrees on right
28	1 m, 50° degrees on right
29	1 m, outer edge on right
30	1.5 m, outer edge on left
31	1.5 m, 45° degrees on left
32	1.5 m, 40° degrees on left
33	1.5 m, 30° degrees on left
34	1.5 m, 20° degrees on left
35	1.5 m, 10° degrees on left
36	1.5 m centre
37	1.5 m, 10° degrees on right
38	1.5 m, 20° degrees on right
39	1.5 m, 30° degrees on right
40	1.5 m, 40° degrees on right
41	1.5 m, 45° degrees on right
42	1.5 m, outer edge on right
43	2 m, outer edge on left
44	2 m, 40° degrees on left
45	2 m, 30° degrees on left
46	2 m, 20° degrees on left
47	2 m, 10° degrees on left
48	2 m centre
49	2 m, 10° degrees on right
50	2 m, 20° degrees on right
51	2 m, 30° degrees on right
52	2 m, 40° degrees on right
53	2 m, outer edge on right
54	2.5 m, outer edge on left
55	2.5 m, 30° degrees on left
56	2.5 m, 20° degrees on left
57	2.5 m, 10° degrees on left
58	2.5 m centre
59	2.5 m, 10° degrees on right
60	2.5 m, 20° degrees on right
61	2.5 m, 30° degrees on right
62	2.5 m, outer edge on right
63	3 m, outer edge on left
64	3 m, 30° degrees on left
65	3 m, 20° degrees on left
66	3 m, 10° degrees on left
67	3 m centre
68	3 m, 10° degrees on right
69	3 m, 20° degrees on right
70	3 m, 30° degrees on right
71	3 m, outer edge on right
72	3.5 m, outer edge on left
73	3.5 m, 20° degrees on left
74	3.5 m, 10° degrees on left
75	3.5 m centre
76	3.5 m, 10° degrees on right
77	3.5 m, 20° degrees on right
78	3.5 m, outer edge on right
79	4 m outer edge on left
80	4 m centre
81	4 m outer edge on
82	Middle head height 4 m

**Table 3 microorganisms-10-02241-t003:** Viral transfer per wipe.

Sample Position	Number of Virus per Wipe (Average of Two Runs)
Scenario 1: Single cough to hand	375
Scenario 2: Rail after hand transfer	75
Scenario 2: Hand after contact with the rail	50
Scenario 3: Single cough to a bare inner elbow	212.5
Scenario 4: Hand after contact with contaminated inner elbow	25
Scenario 4: Rail after contact with the hand that touched the inner elbow	25
Scenario 4: Bare inner elbow after contact with a hand	None detected
Scenario 5: Single cough to a sleeved inner elbow	37.5
Scenario 6: Hand after contact with a sleeved contaminated inner elbow	None detected
Scenario 6: Rail after contact with the hand that touched the sleeved inner elbow	None detected
Scenario 6: A sleeved inner elbow after contact with a hand	None detected

## Data Availability

Data is contained within the article, but additional data and images are available on request.

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
