# Peer review of "Simulating the Environmental Spread of SARS-CoV-2 via Cough and the Effect of Personal Mitigations"

_microorganisms, 2022, doi:10.3390/microorganisms10112241_

Round 1

Reviewer 1 Report

The research was well conducted and well described. A nice model was used to study the spread of virusses, but a few photographs of a 'patient' couching into the hand and elbow would make it more palatable. Then the readership could see whether the model indeed mimics the spread of a patient coughing. Now it seems that the model does not block the spread in the same way as a person would do. The mouth is not covered the same way a person would do. it seems that it covers it less (which also might the case in some individues).

Author Response

Thank you for your comments. 

I have added a few photographs of people coughing at the same distance to those used within the model.

The photograph angles do make it appear that the hand and elbow were further away from the mouth than a person would normally do. I have clarified within the text that the hand and elbow were position close but not touching the mouth/cough origin. We did not measure the distance but it was approximately a thumbs width (2cm) from the mouth. During initial observations it was observed that many people covered their mouth but did not touch their face or press their mouth into their elbow. I think the addition of the patient photos should clarify things.

Reviewer 2 Report

The manuscript describes a pilot study using a cough simulator to assess whether coughing into a hand or an elbow reduces the amount of aerosolized virus. The study appears to be scientifically sound and the results are presented in a clear manner. I note one minor typo in the manuscript: there is a 'Pla' in front of one of the Scenario 4 in Table 3. 

Author Response

Thank you for pointing out the typo. I have amended the final version.